# Reusable Options through Gradient-based Meta Learning

**David Kuric**                                                                                          *d.kuric@uva.nl*
*AMLab, University of Amsterdam*

**Herke van Hoof**                                                                                     *h.c.vanhoof@uva.nl*
*AMLab, University of Amsterdam*

**Reviewed on OpenReview:** *https://openreview.net/forum?id=qdDmxzGuzu*

## Abstract

Hierarchical methods in reinforcement learning have the potential to reduce the amount of decisions that the agent needs to perform when learning new tasks. However, finding reusable useful temporal abstractions that facilitate fast learning remains a challenging problem. Recently, several deep learning approaches were proposed to learn such temporal abstractions in the form of options in an end-to-end manner. In this work, we point out several shortcomings of these methods and discuss their potential negative consequences. Subsequently, we formulate the desiderata for reusable options and use these to frame the problem of learning options as a gradient-based meta-learning problem. This allows us to formulate an objective that explicitly incentivizes options which allow a higher-level decision maker to adjust in few steps to different tasks. Experimentally, we show that our method is able to learn transferable components which accelerate learning and performs better than existing prior methods developed for this setting. Additionally, we perform ablations to quantify the impact of using gradient-based meta-learning as well as other proposed changes.

## 1 Introduction

In hierarchical reinforcement learning (HRL), an agent uses various levels of abstraction to simplify learning in large or complex environments. This is commonly achieved by utilizing a multi-level policy which can be broken up into making high-level decisions (e.g., which out of several low level controllers should be active) and low-level decisions (which actions should a low level controller take). Many such approaches utilize temporal abstraction, ie., the notion that high-level actions are only chosen at certain time steps and then continue to influence the decision making over multiple time steps in the future. A clear benefit of utilizing such high-level decisions is the abstraction of the problem to a simpler one and consequently, easier learning. For example, consider a navigation problem where the agent needs to travel to a city on the other side of the country. In this problem, it is much easier to learn which route to take if the agent considers a few high-level decisions: sequence of cities, rather than low-level actions: sequence of intersections. However, when solving this problem with high-level actions, a mapping from high-level actions to corresponding sequence of low-level actions that move the agent from one city to another needs to be available. This mapping can be provided as a form a prior knowledge, but in most settings this information is not available. Therefore, it is often necessary to learn such mapping from experience and then reuse it when learning similar new tasks.

In order to study these types of problems in the context of reinforcement learning, one can utilize the options framework (Sutton et al., 1999). In this framework, a hierarchical policy is composed of options (modules that encapsulate low-level sub-policies), and a high-level policy that chooses among them. Options have their own termination function, and control is given back to the high-level policy only when the earlier option terminates. Therefore, options define temporally extended behaviors through their sub-policies and can be seen as a mapping from high-level actions to a sequence of low-level actions.

Recent prior work on options proposed a way to learn the options in an end to end manner from experience with policy gradient methods (Bacon et al., 2017; Smith et al., 2018). These methods learned the options in

the context of a single task. However, options learned by single-task algorithms are not incentivized to be reusable, and thus often end up too task-specific to be used in related tasks (Harb et al., 2018).

To overcome this issue, the followup work by Frans et al. (2018) proposed to learn shared options in multiple tasks by alternating between fine-tuning of the high-level policy to the sampled task and joint optimization of both high-level policy and options. While this optimization scheme worked well in practice, it does not pass any gradient signal between high-level policy and options during the training. Furthermore, even though the options produced by this method were more suited for the subsequent learning of new tasks, it used options that terminate after fixed number of low-level actions. Such adjustment simplifies the learning problem because it fixes a termination function. However, the need to define a shared option length can be prohibitive because different suitable high-level actions may have very different lengths. Additionally, these lengths can also depend on the randomness in the environment. In such cases, the length of high-level actions cannot be captured with a single hyperparameter. Moreover, the value of this hyperparameter is not learned and thus requires some prior knowledge about the tasks.

In this work, we identify the shortcomings of current methods and their ability to learn shared reusable options from multiple tasks in a setting where limited to no prior knowledge about tasks is available. We discuss the desired properties of such shared reusable options and use these properties to formulate this learning problem as a concrete meta-learning problem. This formulation then allows us to connect this problem to gradient-based meta-learning and to propose a method that adapts a well-known meta-learning algorithm (Finn et al., 2017) to learn reusable options from multiple tasks. This new method provides a more principled alternative to optimize the meta-learning objective when compared to prior work developed for this setting (Frans et al., 2018). Additionally, it allows one to use state-dependent terminations and learn multiple options from the same experience. In our experiments, we empirically verify that both gradient-based meta-learning and learned terminations contribute to learning reusable options that achieve better performance when applied on new unseen tasks from the same domain.

## 2 Background and Notation

We will consider environments which are episodic Markov decision processes (MDPs). An MDP $\mathcal{M}$ is a tuple $\langle \mathbb{S}, \mathbb{A}, p_0, p_s, R, \gamma \rangle$ with $\mathbb{S}$ being a set of states, $\mathbb{A}$ a set of actions, $p_0(\boldsymbol{s}_0)$ a probability distribution of initial states, $p_s(\boldsymbol{s}'|\boldsymbol{s}, \boldsymbol{a})$ a transition probability distribution, $R(\boldsymbol{s}, \boldsymbol{a})$ a reward function and $\gamma$ a discount factor.

An agent with a stochastic policy $\pi$ interacts with an environment in the following way. At every timestep $t$, the agent receives a state of the environment $\boldsymbol{s}_t \in \mathbb{S}$ and selects an action $\boldsymbol{a}_t \in \mathbb{A}$ according to a policy $\pi(\boldsymbol{a}_t|\boldsymbol{s}_t, \boldsymbol{\theta})$ parametrized by $\boldsymbol{\theta}$. Depending on the current state and the action performed, the environment provides the agent with a new state $\boldsymbol{s}_{t+1} \sim p_s(\boldsymbol{s}_{t+1}|\boldsymbol{s}_t, \boldsymbol{a}_t)$ and a scalar reward $r_t = R(\boldsymbol{s}_t, \boldsymbol{a}_t)$. This process is repeated until a so-called terminal state is reached. We define a trajectory $\tau$ as an ordered sequence of all states actions and rewards in a single episode $\tau = (\boldsymbol{s}_0, \boldsymbol{a}_0, r_0, ..., \boldsymbol{s}_T)$. Similarly, the history at timestep $t$ consists of all states and actions preceding $\boldsymbol{a}_t$, $\boldsymbol{h}_t = (\boldsymbol{s}_0, \boldsymbol{a}_0, ..., \boldsymbol{s}_t)$. The state value function is defined as $V^\pi(\boldsymbol{s}) = \mathbb{E}_{\tau \sim p(\tau|\pi)}[G_t(\tau)|\boldsymbol{s}_t = \boldsymbol{s}]$ where the discounted return at timestep $t$ is defined as $G_t(\tau) = \sum_{t'=t}^{T} \gamma^{(t'-t)} r_{t'}$. The agent's objective is to maximize the expected discounted return $J = \mathbb{E}_{\tau \sim p(\tau|\pi)}[G_0(\tau)]$. We can maximize the objective with gradient descent by estimating the policy gradient $\nabla_{\boldsymbol{\theta}} J \approx \mathbb{E}_{\tau \sim p(\tau|\pi(\boldsymbol{\theta}))}[\sum_{t=0}^{T} \nabla_\theta \log \pi(\boldsymbol{a}_t|\boldsymbol{s}_t, \boldsymbol{\theta}) A^\pi(\boldsymbol{s}_t, \boldsymbol{a}_t)]$ using Monte Carlo sampling, where $A^\pi$ is an advantage estimator such as the generalized advantage estimator $A^{GAE}$ (Schulman et al., 2015).

### 2.1 Options

The options framework (Sutton et al., 1999) is a framework for temporal abstraction in reinforcement learning that consists of options $\omega = \langle \mathbb{I}^\omega, \pi^\omega, \xi^\omega \rangle$ and a policy over options $\pi^\Omega(\omega|\boldsymbol{s}, \boldsymbol{\theta}_\Omega)$. Each option $\omega$ has an initiation set, a sub-policy and a termination function. The initiation set $\mathbb{I}^\omega$ is a set of states in which an option can be selected (initiated) and in our case it is the whole state space ($\mathbb{I}^\omega = \mathbb{S}$). A sub-policy $\pi^\omega(\boldsymbol{a}|\boldsymbol{s}, \boldsymbol{\theta}_\omega)$, also called low-level policy, is a regular policy that acts in the environment. Lastly, the termination function $\xi^\omega(\boldsymbol{s}, \boldsymbol{\theta}_\xi)$ is a function that outputs the probability of the termination for option $\omega$ in a given state. We denote the parameters of policy over options, sub-policies and termination functions as $\boldsymbol{\theta}_\Omega$, $\boldsymbol{\theta}_\omega$ and $\boldsymbol{\theta}_\xi$ respectively.

In our work, we use the Inferred Option Policy Gradient (IOPG, Smith et al. (2018)), a recently introduced policy gradient method for learning options that treats options as latent variables during gradient calculation. This method allows for updating all options based on their responsibilities for a given action which results in better data efficiency. The IOPG gradient can be calculated from sampled trajectories:

$$\nabla_{\boldsymbol{\theta}} J \approx \mathbb{E}_{\tau \sim p(\tau | \pi(\boldsymbol{\theta}))} \left[ \sum_{t=0}^{T} \nabla_{\boldsymbol{\theta}} \log \pi(\boldsymbol{a}_t | \boldsymbol{h}_t, \boldsymbol{\theta}) A^{\pi}(\boldsymbol{s}_t, \boldsymbol{a}_t) \right] \over \underbrace{\sum_{\omega^i} p(\omega_t^i | \boldsymbol{s}_{[0:t]}, \boldsymbol{a}_{[0:t-1]}, \boldsymbol{\theta}) \pi^{\omega^i}(\boldsymbol{a}_t | \boldsymbol{s}_t, \boldsymbol{\theta}_{\omega})},$$

where $\pi(\boldsymbol{a}_t | \boldsymbol{h}_t, \boldsymbol{\theta})$ can be decomposed as indicated in Equation 1 and we use $\boldsymbol{\theta}$ to denote the concatenation of all parameters $(\boldsymbol{\theta}_{\Omega}, \boldsymbol{\theta}_{\omega}, \boldsymbol{\theta}_{\xi})$. The probability $p(\omega_t^i | \boldsymbol{s}_{[0:t]}, \boldsymbol{a}_{[0:t-1]}, \boldsymbol{\theta})$ represents the probability that option $\omega^i$ was active at timestep $t$ given past actions and states and can be computed recursively:

$$p(\omega_{t+1}^j | \boldsymbol{s}_{[0:t+1]}, \boldsymbol{a}_{[0:t]}, \boldsymbol{\theta}) = \frac{\sum_{\omega^i} p(\omega_t^i | \boldsymbol{s}_{[0:t]}, \boldsymbol{a}_{[0:t-1]}, \boldsymbol{\theta}) \pi^{\omega^i}(\boldsymbol{a}_t | \boldsymbol{s}_t, \boldsymbol{\theta}_{\omega}) \tilde{\pi}^{\Omega}(\omega_{t+1}^j | \omega_t^i, \boldsymbol{s}_{t+1}, \boldsymbol{\theta}_{\Omega}, \boldsymbol{\theta}_{\xi})}{\sum_{\omega^k} p(\omega_t^k | \boldsymbol{s}_{[0:t]}, \boldsymbol{a}_{[0:t-1]}, \boldsymbol{\theta}) \pi^{\omega^k}(\boldsymbol{a}_t | \boldsymbol{s}_t, \boldsymbol{\theta}_{\omega})}, \tag{2}$$

with option transition probability $\tilde{\pi}^{\Omega}(\omega_{t+1}^j | \omega_t^i, \boldsymbol{s}_{t+1}, \boldsymbol{\theta}_{\Omega}, \boldsymbol{\theta}_{\xi})$ given by:

$$\tilde{\pi}^{\Omega}(\omega_{t+1}^j | \omega_t^i, \boldsymbol{s}_{t+1}, \boldsymbol{\theta}_{\Omega}, \boldsymbol{\theta}_{\xi}) = \xi^{\omega^i}(\boldsymbol{s}_{t+1}, \boldsymbol{\theta}_{\xi}) \pi^{\Omega}(\omega^j | \boldsymbol{s}_{t+1}, \boldsymbol{\theta}_{\Omega}) + \mathbf{1}_{\omega^j = \omega^i} \left[ 1 - \xi^{\omega^i}(\boldsymbol{s}_{t+1}, \boldsymbol{\theta}_{\xi}) \right]. \tag{3}$$

## 2.2 Model-Agnostic Meta-Learning (MAML)

Model-Agnostic Meta-Learning (Finn et al., 2017) is a meta-learning technique that trains a model for maximum post-adaptation performance on a distribution of tasks $\mathcal{M} \sim p(\mathcal{M})$. The adaptation consists of one or several inner gradient updates. If we consider an estimator $f_{\theta}$ with parameters $\theta$ and a task-specific loss $\mathcal{L}^{\mathcal{M}}$, a supervised learning objective with a single inner update can be formalized as shown in Equation 4. In order to optimize this objective one only needs to take a gradient of this expression. This can be easily achieved with automatic differentiation frameworks by backpropagating through the gradient update.

$$\min_{\boldsymbol{\theta}} \mathbb{E}_{\mathcal{M} \sim p(\mathcal{M})} \left[ \mathcal{L}^{\mathcal{M}} \left( f_{\boldsymbol{\theta} - \alpha \nabla_{\boldsymbol{\theta}} \mathcal{L}^{\mathcal{M}}(f_{\boldsymbol{\theta}})} \right) \right] \tag{4}$$

One can similarly use this approach with a reinforcement learning objective:

$$\max_{\boldsymbol{\theta}} \mathbb{E}_{\mathcal{M} \sim p(\mathcal{M})} \left[ \mathbb{E}_{\tau' \sim p\left(\tau' | \pi\left(\boldsymbol{\theta} + \alpha \nabla_{\boldsymbol{\theta}} \mathbb{E}_{\tau \sim p(\tau | \pi(\boldsymbol{\theta}))} [G_0^{\mathcal{M}}(\tau)]\right)\right)} \left[ G_0^{\mathcal{M}}(\tau') \right] \right] \tag{5}$$

However, the implementation with an automatic differentiation framework differs because a simple backpropagation through the computation graph of the gradient produces biased gradients (Al-Shedivat et al., 2018; Stadie et al., 2018). This is due to an additional dependency of the sampling distribution on parameters that is not present in the supervised learning objective. To produce correct higher order gradients with automatic differentiation frameworks we utilize an objective with DiCE[1] (Foerster et al., 2018; Farquhar et al., 2019).

## 3 Reusable Options

Despite extensive research on options, there is not yet a single answer to the question: *What is a good option?* Consequently, there is also no consensus on the objective for an option learning method. In this paper we take the view that good options should accelerate the learning of new tasks. In other words, we consider options good if they allow us to achieve high return on new tasks as fast as possible. We assume that we are given a distribution of relevant training tasks $p(\mathcal{M})$ with similar (hierarchical) structure but different reward or transition functions. Our goal is then to learn options that maximize expected return on this task distribution as well as the expected return achieved on unseen tasks from the same task distribution.

---

[1]For more details about DiCE see Appendix A.

One could formulate this as maximizing $\mathbb{E}_{\mathcal{M} \sim p(\mathcal{M})} \left[ \mathbb{E}_{\tau \sim p(\tau | \pi(\boldsymbol{\theta}))} \left[ G^{\mathcal{M}}(\tau) \right] \right]$ where trajectories $\tau$ are generated using a hierarchical policy with options parametrized by $\boldsymbol{\theta}$. However, maximizing immediate return with such objective directly is not beneficial because it learns a single fixed policy that solves multiple tasks. The solutions to these tasks can have conflicting requirements as shown in Figure 1, where some tasks require different decisions in certain states. Consequently, a single trained policy would likely be a compromise that performs reasonably well, but sub-optimally, on each individual task. Since the learning of a hierarchical policy that would lead to good zero-shot performance is not feasible, we instead aim for shared options that capture reusable sub-behaviors. Such options should achieve good performance after the policy over options adapts to the task at hand. This objective can be formulated as:

$$\max_{\boldsymbol{\theta}_\omega, \boldsymbol{\theta}_\xi} \mathbb{E}_{\mathcal{M} \sim p(\mathcal{M})} \left[ \mathbb{E}_{\tau \sim p\left(\tau | \pi(\boldsymbol{\theta}_\omega, \boldsymbol{\theta}_\xi, \boldsymbol{\theta}_\Omega^*)\right)} \left[ G_0^{\mathcal{M}}(\tau) \right] \right] \tag{6}$$

$$\boldsymbol{\theta}_\Omega^* = \arg\max_{\boldsymbol{\theta}_\Omega} \mathbb{E}_{\tau \sim p(\tau | \pi(\boldsymbol{\theta}_\omega, \boldsymbol{\theta}_\xi, \boldsymbol{\theta}_\Omega))} \left[ G_0^{\mathcal{M}}(\tau) \right], \tag{7}$$

where $\boldsymbol{\theta}_\Omega^*$ is a high-level policy adapted to the specific task $\mathcal{M}$. While one could also perform maximization over all parameters in Equations 6 and 7, such an approach would not enforce options that capture reusable sub-behaviors. This is because the algorithm can choose to not change the high-level policy and instead make a large change in sub-policies during the adaptation. Nevertheless, we consider this variant in our experiments as an ablation.

The two-level optimization problem in Equations 6 and 7 consist of inner maximization over the parameters of the high-level policy $\boldsymbol{\theta}_\Omega$, which are used to select options, and outer maximization over option parameters $\boldsymbol{\theta}_\omega, \boldsymbol{\theta}_\xi$. Consequently, it is not trivial to optimize with commonly used methods such as the gradient descent because of two reasons. Firstly, in order to get the gradient of the outer objective, one needs to differentiate through the inner maximization in Equation 7 and secondly, the inner optimization problem is not trivial to solve and may require tens or hundreds of gradient steps to optimize. However, we can nullify the latter by simplifying the problem and restricting the number of inner adaptation gradient steps to a fixed number $L$:

$$\max_{\boldsymbol{\theta}_\omega, \boldsymbol{\theta}_\xi} \mathbb{E}_{\mathcal{M} \sim p(\mathcal{M})} \left[ \mathbb{E}_{\tau^L \sim p\left(\tau^L | \pi(\boldsymbol{\theta}_\omega, \boldsymbol{\theta}_\xi, \boldsymbol{\theta}_\Omega^L)\right)} \left[ G_0^{\mathcal{M}}(\tau) \right] \right] \tag{8}$$

$$\boldsymbol{\theta}_\Omega^{j+1} = \boldsymbol{\theta}_\Omega^j + \alpha_{in} \nabla_{\boldsymbol{\theta}_\Omega^j} \mathbb{E}_{\tau^j \sim p\left(\tau^j | \pi(\boldsymbol{\theta}_\omega, \boldsymbol{\theta}_\xi, \boldsymbol{\theta}_\Omega^j)\right)} \left[ G_0^{\mathcal{M}}(\tau^j) \right]. \tag{9}$$

This adjustment also fixes the differentiability of the inner optimization problem because Equations 8 and 9 are conceptually similar to MAML applied to reinforcement learning (Equation 5). The two differences with the standard MAML are multiple inner updates $L$ and limiting inner optimization to a subset of all parameters $\boldsymbol{\theta}_\Omega$. Such adjustments are well supported in the MAML framework. Finn et al. (2017) used multiple inner updates and as was demonstrated by Zintgraf et al. (2019) and Antoniou et al. (2019), reducing the number of parameters that are tuned in the inner updates can be beneficial and lead to better performance. Consequently, we propose to use this MAML-like objective with a policy gradient method to learn reusable options. Note that it is also possible to learn the initial high-level policy parameters by performing the maximization over $\boldsymbol{\theta}_\Omega$ in Equation 8. However, we chose fixed uniform initial high-level policy to facilitate exploration during both training and test phases and to encourage more uniform option usage. Additionally, it was shown that fixed initialization of inner loop MAML parameters in supervised learning setting makes the algorithm more robust to the choice of inner learning rate (Zintgraf et al., 2019).

Another important questions from the options research are: *How many options should one use?* and *When should a termination occur?*. Although these do not seem to be connected at first glance, we show that in the task of learning options from multiple environments, there is a link between these two questions. This link is demonstrated by the example in Figure 1. In this example, we consider four tasks with the same starting location but different goals. The optimal trajectories for each task lead from the starting location straight to the goal. We assume that learned options should be as long as possible to simplify the optimization of high-level policy when faced with a task. This assumption is based on the premise that learning problems that contain fewer decisions (high-level actions) should be easier to solve. Consequently, the adaptation to new tasks with longer options should also be faster as long as some of the learned sub-behaviors can be used in new tasks.

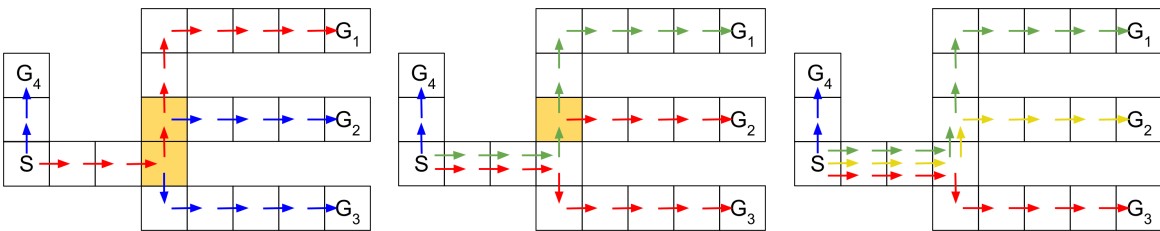

Figure 1: Example in which optimal trajectories in multiple tasks cannot be represented with a single shared policy. The maximum length of options that can represent such optimal trajectories changes with number of options. Options are shown in different color. Necessary terminations are depicted as a filled square.

If we set the number of options equal to the number of tasks, each sub-policy can learn to solve one task and there is no need to combine multiple options. Therefore, the longest options are options without terminations. However, once we start to decrease the number of options, parts of trajectories have to be reused to represent optimal trajectories of 4 tasks with only 3 options. This is demonstrated in the middle of Figure 1, where only one option needs to terminate after several timesteps. Similarly, when using only 2 options, more terminations are needed and the options tend to become shorter. In addition to showing the connection between the amount and positions of desirable terminations and number of options, this example also highlights why choosing a hyperparameter that represents predetermined length of options, as done in a number of prior approaches (Frans et al., 2018; Li et al., 2020), can be challenging. This is because the appropriate hyperparameter value can change based on the number of available options and is usually not known apriori. Additionally, such an approach would most likely not work well in environments where movement actions can fail with certain probability because agent could arrive into a state in which it needs to terminate after taking different amount of steps.

These two examples demonstrate why using terminations that depend on current state as proposed in the original options framework work (Sutton et al., 1999) may be preferred. To this end we propose to learn reusable options (including terminations) from multiple tasks by combining the Inferred Option Policy Gradient (IOPG) with the adapted version of MAML. Similar to prior work, we assume that number of options is given as a hyperparameter. The objective we propose in Equations 8 and 9 then implicitly discourages pathological options. For example, the scenario where only a single option is ever used (option collapse) is discouraged as the agent that only uses a single option to solve multiple tasks would achieve lower average return. Similarly, the scenario where options terminate every time step is discouraged because by optimizing for fast adaptation of high-level policy, the algorithm is incentivized to use options that are not too short (degenerate 1-step options). This is because options that are too short would not allow high-level policy to fully adapt to new tasks within several inner updates and would also lead to lower average return.

IOPG also allows us to learn terminations and to update all options at the same time based on their responsibilities, i.e., the probability that the option was active given the history $\boldsymbol{h}_t$ of states and actions so far. This can lead to better data-efficiency when compared to other methods that only update a single option at a time, such as the Option Critic (Bacon et al., 2017), but comes at the cost of increased computation time. Additionally, IOPG is also not reliant on a learned option-value function which is helpful in the meta learning setting because the option-value function changes after every inner update. This is in contrast with the Option Critic and several recent methods that extend it (Khetarpal et al., 2020; Klissarov & Precup, 2021). Combining these methods with MAML would thus be difficult. Additionally, there is no straightforward way to apply these algorithms in a setting with multiple tasks when the agent does not know which environment it is in.

### 3.1 Fast Adaptation of Modular Policies

The resulting algorithm for Fast Adaptation of Modular Policies (FAMP) is outlined in Algorithm 1. After $L$ inner updates, the gradient of the objective with respect to the outer parameters is calculated. In principle, we would like to optimize for performance after a moderate number of gradient updates $L$ such as 10 or 20. However, with more inner updates the resulting gradient of the objective becomes noisier due to the usage

---

**Algorithm 1:** Fast Adaptation of Modular Policies (FAMP)

---

**input** : A distribution of tasks $p(\mathcal{M})$, number of options $N$, env samples per update $M$, adaptation
steps $L$, episodes per update $k$, learning rates $\alpha_{in}, \alpha_{out}$

**1** randomly initialize $\boldsymbol{\theta}_\Omega, \boldsymbol{\theta}_\xi, \boldsymbol{\theta}_\omega$

**2** **while** *not converged* **do**

**3**    $\boldsymbol{g}_{\langle\boldsymbol{\theta}_\xi,\boldsymbol{\theta}_\omega\rangle} = 0$         $\triangleright$ zero accumulated gradients

**4**    **for** env $\leftarrow 1$ **to** $M$ **do**

**5**      sample task $\mathcal{M} \sim p(\mathcal{M})$

**6**      set $\boldsymbol{\theta}'_\Omega = \boldsymbol{\theta}_\Omega$

**7**      **for** $l \leftarrow 1$ **to** $L+1$ **do**

**8**        sample $k$ episodes $\tau_{1:k}$ on $\mathcal{M}$ using $\pi\left(\boldsymbol{\theta}'_\Omega, \boldsymbol{\theta}_\omega, \boldsymbol{\theta}_\xi\right)$

**9**        fit a baseline using data from $\tau_{1:k}$

**10**       compute $A^{GAE}$ and $\log \pi(\boldsymbol{a}_t | \boldsymbol{h}_t, \boldsymbol{\theta}'_\Omega, \boldsymbol{\theta}_\omega, \boldsymbol{\theta}_\xi)$ for all $\tau_{1:k}$

**11**       compute loss $J$ according to Equation 1

**12**       **if** $l < L+1$ **then**

**13**         $\boldsymbol{\theta}'_\Omega = \boldsymbol{\theta}'_\Omega + \alpha_{in}\nabla_{\boldsymbol{\theta}'_\Omega} J$        $\triangleright$ adapt $\pi^\Omega$

**14**       **else**

**15**         $\boldsymbol{g}_{\langle\boldsymbol{\theta}_\xi,\boldsymbol{\theta}_\omega\rangle} = \boldsymbol{g}_{\langle\boldsymbol{\theta}_\xi,\boldsymbol{\theta}_\omega\rangle} + \nabla_{\langle\boldsymbol{\theta}_\xi,\boldsymbol{\theta}_\omega\rangle} J$    $\triangleright$ accumulate gradients of outer objective

**16**       **end**

**17**      **end**

**18**    **end**

**19**    $\langle\boldsymbol{\theta}_\xi, \boldsymbol{\theta}_\omega\rangle = \langle\boldsymbol{\theta}_\xi, \boldsymbol{\theta}_\omega\rangle + \alpha_{out}\frac{1}{N}\boldsymbol{g}_{\langle\boldsymbol{\theta}_\xi,\boldsymbol{\theta}_\omega\rangle}$        $\triangleright$ update options

**20** **end**

**output:** initial policy over options $\pi^\Omega(\omega|\boldsymbol{s}, \boldsymbol{\theta}_\Omega)$, sub-policies $\pi^\omega(\boldsymbol{a}|\boldsymbol{s}, \boldsymbol{\theta}_\omega)$, terminations $\xi^\omega(\boldsymbol{s}, \boldsymbol{\theta}_\xi)$

---

of Monte Carlo estimates in each inner update. Furthermore, the time complexity of gradient computation and sample complexity both scale linearly with the number of inner updates. In practice we found a range from 2 to 4 update steps to be acceptable. A benefit of gradient-based meta-learning is that even though the model is optimized for performance after $L$ adaptation steps, it can still be improved after $L$ updates by performing more steps of gradient descent. An important design choice is the state value function estimator. In the MAML RL setting the policy constantly changes in every inner update. It is thus difficult to use past trajectories for fitting the value function. Furthermore, a different value function needs to be computed for each task. We therefore use a linear time-state dependent baseline (Duan et al., 2016a), which works better than more complex baselines with little data.

## 4 Experiments

In order to evaluate our method empirically and show the benefits of learned terminations as well as end-to-end learning with gradient-based meta-learning, we perform experiments in Taxi and AntMaze domains [2]. In these experiments, we compare to the closest state of the art method and perform ablations to highlight the effects of different objectives and learned terminations on the final performance. In both domains, we are interested in the final average performance on the set of test tasks. Single-task algorithms are trained on these tasks directly. On the other hand, meta-learning algorithms, including our own, are first pre-trained on a set of different but related training tasks until their performance stops improving or they reach one week of training time. We then plot the performance as the function of the number of episodes during the adaptation.

---

[2]Our code is publicly available at `https://github.com/Kuroo/FAMP`.

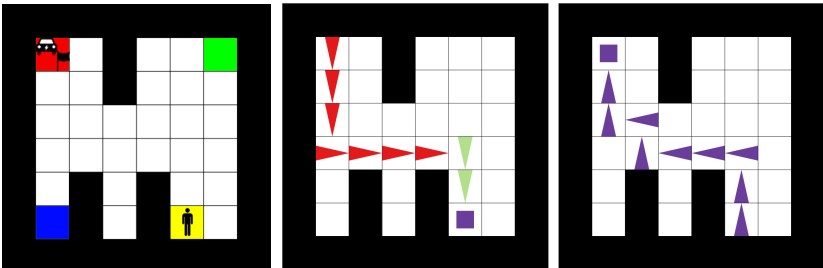

Figure 2: *Left:* Map of a taxi environment with special states and an example task. *Middle and Right:* Visualization of the option usage in this task. Middle part shows states without passenger on board. Right part shows states with passenger. Arrows represent directional actions, pick-up/drop-off is shown as a square. Each action is colored according to the active option.

### 4.1 Taxi

In the first set of experiments we use a modified Taxi environment[3] (Dietterich, 2000). This environment is commonly used in works on hierarchical learning and options (Dietterich, 2000; Igl et al., 2020) and allows one to create many different tasks with shared parts to test the reusability of learned options. Furthermore, subtasks and optimal trajectories have different lengths across tasks and thus may require options with different lengths. Finally, since both state and action space are discrete, the learned options can be visualized and examined. The map of the environment is displayed in Figure 2. An agent acts as a taxi driver who starts in one of the special (colored) locations. Its goal is to pick-up a passenger from one special location and drive him to another special location. Different tasks use different combinations of start, goal and passenger locations. We use 48 combinations as training tasks and 12 as test tasks.

Each task is an MDP in which the agent can use 4 directional actions and two special actions: pick-up/drop-off and no-op. The state space is represented as a one-hot vector that *does not encode any information about the location of the passenger or goal state.* Therefore, in order to facilitate fast adaptation to the (unobservable) passenger and goal locations, the agent must acquire options that can serve as building blocks for exploration. The reward is 2 for reaching the goal and $-0.1$ per step otherwise. Episodes terminate if they take longer than 1500 timesteps. We use a tabular representations for the policy over options, terminations and subpolicies. Our experiments are set up to answer following questions:
**(1) How does FAMP compare to the closest hierarchical method in terms of performance?**
We chose Meta-Learning Shared Hierarchies (MLSH, Frans et al. (2018)) as the strongest meta-learning baseline with options, because it is the closest hierarchical method designed for similar setting in which no extra information about the environment is available. This is in contrast with many other hierarchical and non-hierarchical meta reinforcement learning methods for learning options(Pickett & Barto, 2002; Barreto et al., 2019), which utilize extra information such as the ID of a sampled environment (Mankowitz et al., 2016; Igl et al., 2020; Veeriah et al., 2021). However, unlike our method, MLSH uses time-based terminations and does not use gradient-based meta-learning to optimize its objective. We pre-train MLSH on all training tasks to learn sub-policies. During the test-time we freeze these sub-policies and only allow the high-level policy to adapt to test tasks. This setting is identical to the one used in the original work albeit with a different environment. **(2) How does FAMP compare to a policy that is not trained for fast adaptation and a policy trained from scratch?** We use the *multi-task* baseline to show the performance of a hierarchical policy that optimizes for the immediate average return in multiple tasks rather than the return after several adaptation steps. The whole hierarchical policy of this baseline is first meta-learned and then fine-tuned on test tasks. It corresponds to using Algorithm 1 with $L = 0$ and $\langle \boldsymbol{\theta}_\Omega, \boldsymbol{\theta}_\xi, \boldsymbol{\theta}_\omega \rangle$ instead of $\langle \boldsymbol{\theta}_\xi, \boldsymbol{\theta}_\omega \rangle$ in lines 3, 15 and 19 during training. We also use the *single-task* baseline which learns to solve the test tasks from scratch without any pre-training using IOPG. Therefore, since it does not need to generalize to many tasks and has a flexible policy, we expect it to perform better than other methods after sufficiently long training. However, a meta-learned policy with desirable options should find a good solution quicker. **(3) Is**

---

[3]Details are included in Appendix A.

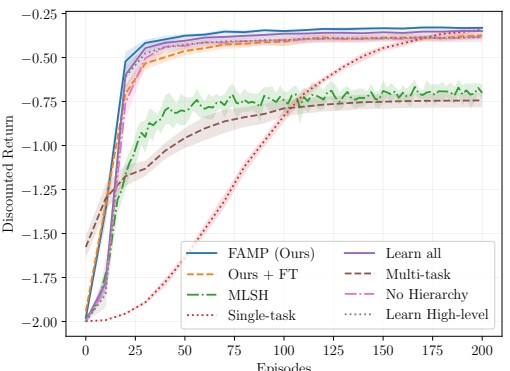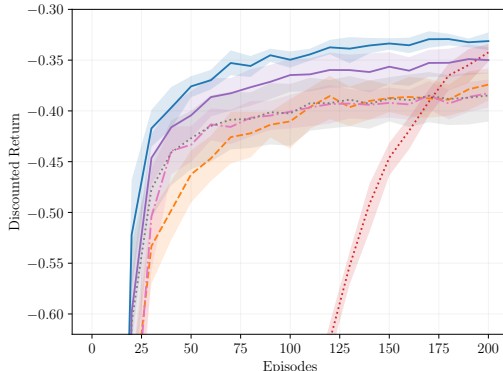

Figure 3: Average performance of different algorithms on Taxi test tasks. Plots show mean and 95% bootstrap confidence intervals over 9 seeds.

**it important to restrict the inner and outer update to specific parts of the hierarchical policy?** To answer this question, we compare the performance of FAMP to *learn high-level* and *learn all* baselines. The former also uses a hierarchical policy with learned terminations but it learns all initial parameters. This corresponds to adjusting line 15 in Algorithm 1 to update all parameters $\langle \boldsymbol{\theta}_\Omega, \boldsymbol{\theta}_\xi, \boldsymbol{\theta}_\omega \rangle$. Similarly, the *learn all* baseline also learns all initial parameters but it also additionally adapts the whole hierarchical policy in the inner update step. This corresponds to adjusting both line 13 and 15 in Algorithm 1 to update all parameters $\langle \boldsymbol{\theta}_\Omega, \boldsymbol{\theta}_\xi, \boldsymbol{\theta}_\omega \rangle$ and can be seen as simply optimizing a hierarchical policy with MAML objective. In addition to the two aforementioned ablations, we use a *no hierarchy* baseline to evaluate the performance of non-hierarchical policy with MAML objective. This can be implemented in practice by using Algorithm 1 with *learn all* adjustments and 1 option. **(4) Are learned terminations important to perform well in this setting?** In order to quantify how important learned terminations are in this setting, we use an ablation of our algorithm with fixed options length as a second baseline (*Ours + FT*). This baseline uses Algorithm 1 to learn the sub-policies but does not learn terminations. Instead, it uses a termination scheme of MLSH where the termination occurs every $c$ steps where $c$ is a hyperparameter. **(5) Does the value of $c$ affect the final performance?** We vary the option length hyperparameter in both MLSH and FAMP to determine whether its value affects the adaptability of learned options and the final performance on test tasks. **(6) How does the number of options and inner updates used during training affect the performance?** We compare the performance of FAMP with different number of options ranging from 2 to 16 and different number of inner update steps $L \in \{1, 2, 3\}$.

**Results**

The performance comparison with baselines is shown in Figure 3. Plots with meta-training curves are included in Appendix B. Our method is able to outperform both MLSH and *multi-task* meta-learning baselines reaching a final performance of $-0.315$. In addition to reaching good final performance, FAMP also outperforms other algorithms in terms of adaptation speed. As expected, the *single-task* baseline eventually overtakes FAMP after more than 200 episodes and reaches a final performance of $-0.284$. However, reaching such performance takes significantly longer than when policies with pre-trained options are used. This demonstrates that FAMP can learn sub-policies and terminations that allow for fast adaptation in similar unseen environments at the cost of slightly lower asymptotic performance. An example trajectory that was produced by the FAMP agent in one of the hardest test tasks is displayed in Figure 2. In this task, the agent is able to combine three options to form an optimal solution.

We further observe in Figure 3 that the *learn all* ablation, which meta-learns all policy parameters, reaches comparable albeit a bit lower performance than FAMP. This highlights the strength of using gradient-based meta-learning in this problem. However, as we have discussed in Section 3, adapting the whole policy usually does not lead to reusable options because a lot of the adaptation is done in sub-policies which may no longer be feasible in more complex tasks. We revisit this issue in the next sub-section. Trained policies of FAMP and *learn all* baseline are available in Appendix B for comparison. Lastly, we observe that the performance

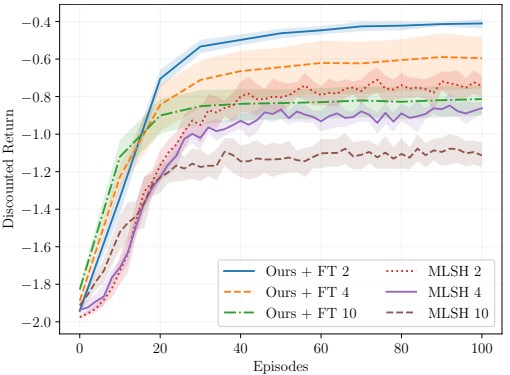 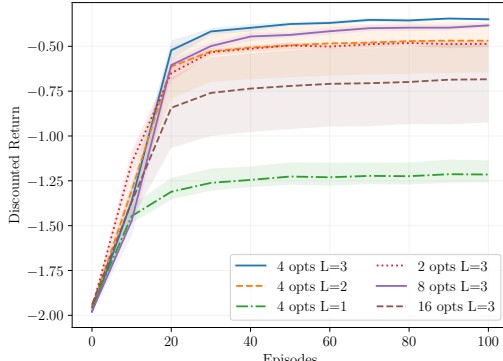

Figure 4: *Left:* Performance of FAMP with fixed terminations and MLSH with terminations after different amount of steps on Taxi test tasks. *Right:* Average performance of our method with different hyperparameter values on Taxi test tasks. Plots show mean and 95% bootstrap confidence intervals over 9 seeds.

of *no hierarchy* and *learn high-level* is lower than when one uses a hierarchical policy and meta-learns options without high-level policy.

In Figure 4 (left), we show the performance of FAMP and MLSH with different option length $c$. We observe that the performance of our algorithm is lower when fixed terminations are used instead of learned terminations. This difference in performance suggests that the termination scheme with fixed amount of steps is too restrictive for this setting. We further observe that FAMP outperforms MLSH across different values of $c$ and both algorithms perform better with smaller option lengths.

In Figure 4 (right), we show how the performance varies with changes to important hyperparameters, namely, the number of options and adaptation steps. We observe that decreasing the number of adaptation steps during training leads to a clear drop in performance especially for $L = 1$. This can be attributed to the policy not being able to switch from exploratory to exploitatory behavior in a single inner update. Unlike the number of adaptation steps, the number of options does not seem to affect the performance much. This highlights the advantage of using inference-based method (IOPG) for learning options since all options are also updated with experience that was generated by other options.

The only noticeable changes in performance are the extremes with 2 or 16 options. A slight drop in performance with 2 options can be explained by noticing that in some states, such as the state 2 squares above the blue special state, one needs to perform 3 different actions to follow the optimal path to red, blue and yellow states. Thus the agent cannot represent the optimal policies with only 2 options. Interestingly, even in this case, the agent is still able to separate trajectories in such a way that it can reach all goals albeit with slightly worse performance. This further illustrates how choosing the appropriate option length a priori can be difficult since it can depend on the number of available options.

We observed that throughout our experiments, the length of options increased with the number of available options. While this is usually a desirable property, increasing the number of options too much can negatively affect reusability of options and may require more episodes per adaptation step in settings with sparser rewards. For example, if the initial high-level policy that is used to select sub-policies is close to uniform and one uses too many options, 10 episodes may be insufficient to try various options before performing a high-level policy update. Determining which option should be selected and appropriately adapting the high-level policy becomes difficult in such cases. Similarly, using too few options can lead to sub-optimal solutions if there are too few options to represent optimal trajectories in all tasks. In such case, the drop in performance may be less or more severe depending on the task.

## 4.2 AntMaze

In the second experiment, we demonstrate the applicability of our method to more complex environments with continuous state and action spaces and perform similar ablations. We use the ant maze domain introduced

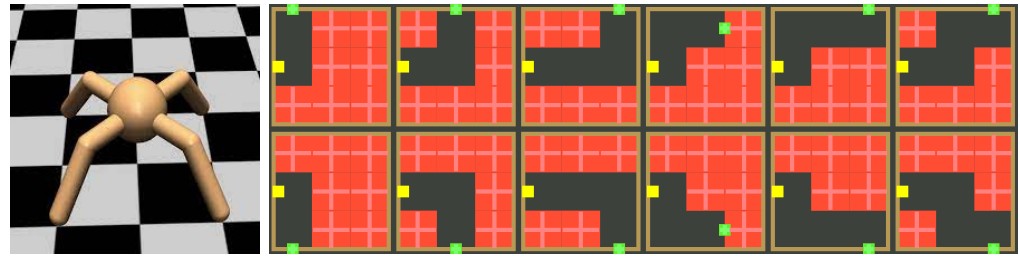

Figure 5: Ant maze tasks. The agent needs to control a simulated 4-legged ant-like robot and move it from yellow square to the green square. Four rightmost tasks are tests tasks.

by Frans et al. (2018) whose tasks are shown in Figure 5. This domain allows us to perform the best possible comparison with MLSH because we can reuse the code and hyperparameter values chosen by the original authors. The only adjustment is removing the orientation resets of the ant, which occurred every 200 steps, because they introduced discontinuities and were not realistic for the robotics scenario they are supposed to imitate. The results of experiments with FAMP and MLSH in the original implementation of the environment were similar and are available in Appendix B.

We use 9 tasks for training and 4 test tasks. In each task the agent needs to move a simulated MuJoCo (Todorov et al., 2012) 4-legged ant-like robot through a small maze towards the goal. Both state space and action space are continuous with 29 and 8 dimensions respectively and the agent receives a positive reward for moving closer to the goal location. However, *states do not contain any information about the maze layout or the location of the goal*. In this settings we consider questions 1-4 from previous environment and verify whether the takeaways from the discrete setting also translate to continuous setting. Similar to the experiments in Taxi domain, we use MLSH as a closest hierarchical baseline, PPO as a single-task baseline, the *learn all* baseline and FAMP with fixed option length (*Ours + FT*). Additionally, we use MAML with a flat policy to compare to an established meta-learning algorithm without options. We used two hidden layers of 64 nodes to represent the components of hierarchical policies and the policy of PPO. For MAML, we increased the layer sizes to 128. In addition to aforementioned baselines, we have also trained an agent with $RL^2$ (Duan et al., 2016b) on these tasks. Because its performance was comparable to MAML, which is another meta-learning algorithm without options that is directly related to our method, we decided to omit $RL^2$ in the plots to keep them uncluttered. Its meta-training curve can be found in Appendix B. For all baselines, we used existing implementations for training and evaluation. Exact hyperparameters and details can be found in Appendix A.

### Results

The comparison of the performance and the speed of adaptation on both train and test tasks can be seen in Figure 6. In both settings, FAMP outperforms the baselines. However, unlike in the previous experiment, FAMP with fixed option length has lower variance and performs slightly better on training tasks. This is likely because in this environment, the terminations do not have to occur at the exact position as was the case in Taxi. Having pre-determined option length thus does not have such adverse effect on the final performance. We have observed similar trends across individual environments (plots are available in Appendix B). Furthermore, we observe that MAML is able to sharply increase its performance during the first few steps of adaptation but is unable to reach the performance of hierarchical methods that utilize options. This suggests that the adaptation of a single policy might not be sufficient to make abrupt enough changes in the agent's behavior that are required to successfully adapt to these tasks. Similarly, meta-learning a hierarchical policy without restricting the adaptation to high-level policy results in low performance. While *learn all* improves its performance during the adaptation, it is the worst of all meta-learning methods. This is in contrast with the simpler environment in which its performance was similar to FAMP. Lastly, we observe that while PPO continuously improves, its performance does not come close to the meta-learning algorithms after 200 episodes. It takes about 1000 episodes to reach the performance of MLSH and if ran sufficiently long, we expect that it would eventually catch up with or even outperform FAMP.

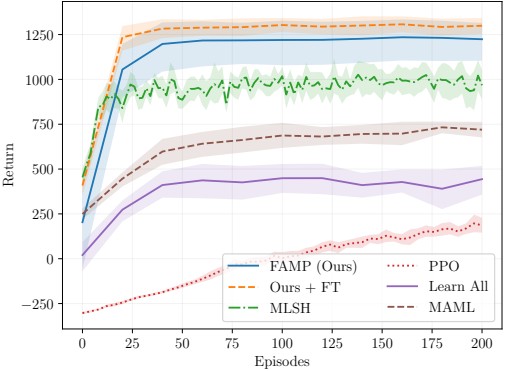 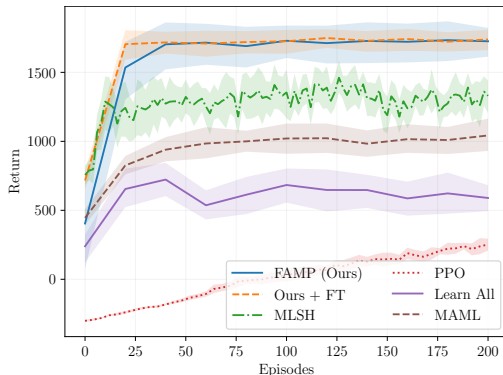

Figure 6: *Left:* Average performance of algorithms on ant training tasks (multi-task setting). *Right:* Average performance of algorithms on ant test tasks (meta-learning setting). Plots show mean and 95% bootstrap confidence intervals over 9 seeds.

While it is not possible to visualize learned options due to the high dimensionality of both state and actions spaces, we can visualize which option is active at each part of the state space. After the high-level policy is fine-tuned, we use the $x$ and $y$ positions of the ant in sampled trajectories to highlight the active option. An example option usage of FAMP in different tasks is shown in Figure 7. In general, the agent uses the cyan option to move upwards and blue to move towards lower parts of the maze while the combination of options is used to fine-tune the direction. This shows that the agent learned a useful abstraction that allows it to perform several different useful behaviors in similar parts of the state space by using meta-trained options and by combining them with a fine-tuned high-level policy.

## 5 Related Work

One of the aims of hierarchical reinforcement learning is to decompose a complex task or policy into simpler units. A thorough overview and classification of existing HRL methods that learn and use such policies can be found in the recent survey by Pateria et al. (2021). Prior approaches include (among others) methods that learn a set of diverse skills (Florensa et al., 2017; Achiam et al., 2018; Eysenbach et al., 2019), or methods that rely on the manager-worker task-decomposition of Feudal Reinforcement Learning (Dayan & Hinton, 1993; Vezhnevets et al., 2017; Nachum et al., 2018). The goal of diversity-based methods is to learn skills that lead to different parts of the state space and behaviors in an unsupervised setting. On the other hand, Feudal Reinforcement Learning methods are guided by the reward function and aim to learn a policy in which each level (manager) provides sub-tasks to the lower level (worker). However, providing sub-goals can be difficult for some tasks (e.g. run in a circle) and manager-worker architecture leads to recursively optimal policies (Dietterich, 2000). This means that in each level the manager is optimal given its workers but the policy as a whole may be sub-optimal. Instead, we can look at the alternative paradigm of the options framework (Sutton et al., 1999).

Several different strategies have been proposed to learn options within this framework. Some works rely on so-called bottleneck states that can be used as sub-goals (McGovern & Barto, 2001; Menache et al., 2002; Niekum & Barto, 2011) whereas others use spectral clustering (Machado et al., 2017). These approaches are particularly suited for settings without reward function but often require prior knowledge about the environment which restricts their applicability. Different from aforementioned methods, end-to-end methods such as the ones which rely on the Option-Critic architecture (Bacon et al., 2017; Riemer et al., 2018; Khetarpal et al., 2020; Klissarov & Precup, 2021) are applicable in more general settings. However, they can be less efficient than concurrently introduced inference-based end-to-end methods (Daniel et al., 2016; Fox et al., 2017; Smith et al., 2018) because they only update the option that generated the action whereas inference-based methods update all options according to their responsibilities for each action. The recently introduced Multi-updates Option Critic (Klissarov & Precup, 2021) is a notable exception since it allows one to update multiple options using the same experience within the Option Critic architecture.

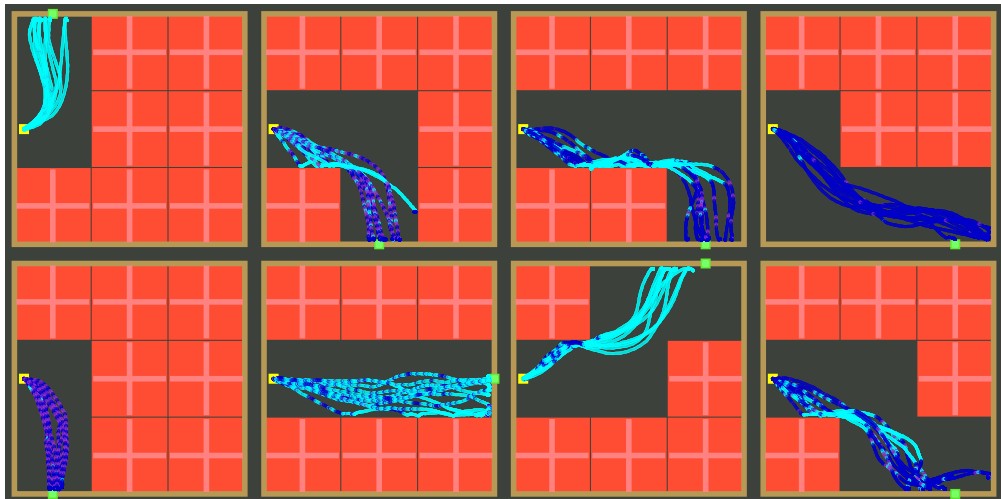

Figure 7: Option usage visualization on ant maze tasks. Plots were created using positions of the ant in trajectories created by adapted policy. Each color represents a different option.

A common problem when using end-to-end methods that learn terminations in a single-task setting is the option collapse (Bacon et al., 2017). It causes the options to terminate after every action or to never terminate. This phenomenon occurs because a single-task can be solved without using options. Consequently, the policy often does not have sufficient incentive to utilize terminations and learn options that are useful for transfer. In such cases the learning of terminations can be facilitated by augmenting the objective with entropy regularization (Smith et al., 2018) or deliberation cost (Harb et al., 2018), regularizing towards a termination prior (Igl et al., 2020), or by using a different objective for terminations (Harutyunyan et al., 2019). Preliminary results also show that the learning of interest functions (Khetarpal et al., 2020) could help to alleviate this issue. In our work, we instead incentivize the algorithm to learn appropriate terminations by training a small amount of options in a multi-task setting. Since the number of available options is lower than the amount of tasks, the agent needs to learn to terminate and combine options in order to solve all tasks. While options were sometimes learned in multi-task setting in prior work (Frans et al., 2018; Li et al., 2020), these works utilized time-based terminations with fixed (Frans et al., 2018) or randomized length (Li et al., 2020). However, as we have discussed in Section 1 and as our ablations have shown, this can lead to problems and degraded performance when options with different lengths are required.

There exist methods which do not employ the techniques mentioned above but still rely on the options framework (Konidaris & Barto, 2007; Zhang & Whiteson, 2019; Li et al., 2020) or task-specific policies (Teh et al., 2017). These approaches often make different assumptions about the tasks and settings in which they are applied and therefore are not directly comparable to our work. Some require policies that solve each environment (Pickett & Barto, 2002) whereas others need environment ID (Mankowitz et al., 2016; Igl et al., 2020) or cumulants that accurately represents task dynamics (Barreto et al., 2019). Close to our work are Adaptive Skills Adaptive Partitions (ASAP) (Mankowitz et al., 2016) and Meta Learning Shared Hierarchies (MLSH) (Frans et al., 2018). ASAP uses a policy gradient method to optimize immediate performance on multiple tasks with known environment ID but does not use neural networks and does not learn terminations. On the other hand, MLSH uses a hierarchical structure with predefined option lengths (time-based terminations) and a problem setting with unknown environment ID. It optimizes for post-adaptation performance by using two alternating phases that either only update the high-level policy or both high-level policy and sub-policies simultaneously. The number of updates in each of these phases is modulated by important task-specific hyperparameters. Because MLSH does not pass the gradient signal between the master and sub-policies, the sub-policies are not explicitly optimized for fast adaptation of high-level policy. This can potentially hinder the adaptability of the final policy and its final performance.

The method we propose instead assumes that policy parameters are updated with gradient descent and aims to learn the parameter values of options. Thus, its problem formulation is closely related to the one of a

recent gradient-based Model-Agnostic Meta-Learning method (Finn et al., 2017), which uses gradient-based meta learning to learn initial parameter values in supervised learning and reinforcement learning settings. The work of Finn et al. (2017) on MAML was extended in followup works that only trained a part of the network (Zintgraf et al., 2019; Raghu et al., 2020) or showed benefits of per-parameter learning rates (Li et al., 2017; Antoniou et al., 2019). Furthermore, several works focused on MAML in a reinforcement learning setting (Al-Shedivat et al., 2018; Stadie et al., 2018; Liu et al., 2019). In particular, Al-Shedivat et al. (2018) and Stadie et al. (2018) pointed out a difference between theory and practical implementation of MAML in automatic differentiation frameworks. This issue was further discussed and resolved in followup works (Foerster et al., 2018; Rothfuss et al., 2019; Farquhar et al., 2019). However, the connection between the learning of reusable options and this type of gradient-based meta-learning has not been explored in prior work.

Instead, a recent work (Veeriah et al., 2021) that utilized both options and meta-learning relied on meta-gradients (Xu et al., 2018; Zheng et al., 2018). Unlike MAML, these have been mainly applied in single-task (single lifetime) settings to learn hyperparameters (Xu et al., 2018; Zahavy et al., 2020) or intrinsic rewards (Zheng et al., 2018; 2020). Consequently, there are some key differences between our work and the work of Veeriah et al. (2021). Firstly, their work meta-learns option reward networks in addition to terminations. Both of these are used to guide the learning of sub-policies and only the high-level policy is trained with a task reward. In our work, all parts of the network are trained using the task reward. Secondly, they let the agent use primitve options that always terminate after a single action and use a switching cost to prevent high-level policy from selecting primitive options too often. Lastly, they use both current state and a task-specific goal encoding as an input to the high-level policy. The agent thus knows which task it is currently in. Their method is thus not applicable in our setting because we consider a setting in which this information is not available.

## 6 Discussion and Future Work

We considered the problem of learning a useful temporal abstraction from multiple tasks in the setting in which there is little prior knowledge available about the environments. Specifically, we were interested in learning options that can accelerate learning in new tasks that are similar to training tasks. We pointed out some of the weaknesses that prior methods for learning of options in similar settings posses. In particular, we have discussed how choosing a fixed options length apriori can be restrictive and detrimental. This is because the ideal value of this hyperparameter can change based on the number of available options. Subsequently, we used a meta-learning formulation of this problem to propose a method that combines the options framework with gradient-based meta-learning and explicitly optimizes learned sub-policies and terminations for performance after several adaptation steps. In our experiments, we have shown that this method outperforms the closest hierarchical and non-hierarchical methods designed for similar meta-learning setting in one discrete and one continuous setting. Finally, we have performed several ablations to better evaluate the benefit of learned terminations and gradient-based meta-learning which distinguish our approach from prior work.

The main limitation of our method is the computationally more expensive calculation of responsibilities in IOPG which cannot be computed in parallel. The computation of responsibilities scales linearly with the amount of timesteps and can become a computation bottleneck during the backpropagation. Alleviating this issue could be a potential direction for future work. Similarly, because IOPG uses policy gradient to update the policy, our method is less data efficient at train time than other methods such as MAML which use TRPO or PPO for their (outer) updates. Extension that would allow for one of the more advanced policy updates to be used in combination with our method could potentially improve the data efficiency. Lastly, while our objective implicitly encourages appropriate terminations, it does not explicitly constrain the number of option terminations. Consequently, learned options may be shorter or longer than intuitively expected as long as they lead to good performance. Future research in this direction could focus on combining our objective with regularization techniques such as deliberation costs to softly encourage more intuitive option lengths.

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
