# OpenReview forum: "Reusable Options through Gradient-based Meta Learning"
_TMLR — Accepted by TMLR_

### Review · Reviewer_shTx · 2023-01-29

**Summary Of Contributions:**

This work proposes a way to obtain re-usable options by framing the problem of option discovery through a meta learning approach. In particular, the authors suggest an objective function in which the meta-policy is updated multiple times before options are themselves updated. This is done with the goal of faster adaptation of the meta-policy. The evaluate their approach, named FAMP, on the classic Taxi domain and a challenging Ant navigation task. In these experiments, FAMP seems to outperform all baselines.

**Audience:**

Yes

**Claims And Evidence:**

Yes

**Requested Changes:**

Running the algorithms for more seeds

Better context for the approach with respect to related work in HRL and meta learning.

Better explanation on the different choices (mentioned above) for the empirical setting.

Clearer discussion on why some baselines are not presented.

**Strengths And Weaknesses:**

**Strengths**
- Builds on well known algorithms (MAML) that perform well in practice
- Evaluation on both a tabular domain and a more challenging continuous control task
- Well-motivated algorithm that investigates most of its choices

**Weaknesses**
- Some of the claims do not reflect recent advances on the options framework
- The general presentation could be improved (e.g. everything around meta gradients and MAML, or the focus on termination probabilities)
- The empirical setup brings many questions (in terms of pertaining and number of seeds in the more challenging domain (only 3!) )


The empirical setting could be greatly improved. First, the fact that only 3 seeds are used on Ant and 5 on the tabular Taxi is a red flag for me. We are constantly suffering from past mistakes in deep RL where practitioners have not been rigorous in their evaluation, and that starts with having a higher number of seeds. Secondly, the amount of pretraining for each is different. In fact, FAMP (the authors method) is trained for one or two orders of magnitude longer than any baselines in the Ant experiment. This is conveniently left out to the appendix, which I find is not a good sign. This should be clearly discussed in the main paper and before substantiated for why this is the case. The main papers brings the argument that each method is trained until it doesn't improve, but is that a fair setting? What is the purpose of this setting, what are we trying to evaluate more precisely? Actually, looking at Figure 8 it even seems that MAML is still learning.

The paper's abstract mentions that several deep learning approaches present shortcomings. From the introduction, this is understood to be the line of work around the option critic (OC) and MLSH. For the former, the problem is option degeneration while for the latter it is the fact that the termination probability is uniformly fixed over states. Concerning the fixed termination probability, a lot of discussion is provided in the paper as to its importance, yet the experiments using deep networks show that there is no significant advantage in learning a termination function. This seems to contradict those paragraphs that mention the important of such a function. Moreover they show that the discussion on the termination function is focused on a case where optimal options can be recovered (i.e. in the tabular case): when options are approximated, their applicability is more flexible due to generalization and this means that we do not need to know exactly when to terminate. Concerning the work on OC, there are more recent results (e.g. Khetarpal et al. 2020) that show its relevance in the context of multi task learning (which is not the setting for the two works referred to in the paper). In fact recent results seem to show that options tend to stabilize in a transfer learning setting. I am not sure if any of the baselines presented would be an equivalent of the OC in this multi task setting.

The paper mentions that IOPG is preferred as OC is on-policy. However, there is recent work that shows a simple extension to OC for learning all options simultaneously called Multi-updates Option Critic (MOC) (Klissarov and Precup, 2021). This work would affect many discussions and claims throughout the paper. It can also potentially affect the experiments, as it is mentioned in the Appendix B that MLSH are more sample efficient since they use PPO, which is the algorithm used by MOC. It could therefore be an important baseline. Another baseline that I would to see is Veeriah et al. 2021, however it is mentioned that this baseline uses privileged information about tasks, can the authors expand on that?

In general I would appreciate either a more precise discussion of the "multi-task" and "learn all" baselines, if not a complete algorithm for each. How can multi-task be doing so badly? This seems to go against recent findings (Kurin et al., 2022)

In the appendix it is said "MAML uses a more complex outer gradient update while FAMP relies on a simple policy gradient." I am not sure I understand what that means exactly, could the authors expand?

Finally, it seems like a discussion about a whole line of work on meta gradients is missing, for example Xu et al 2018, Zheng et al. 2018, or Zahavy et al. 2021. At least some of these works should be cited in my opinion as they are an important part of the literature on meta gradients.


Minor things:

Citation format -> "In particular, (Al-Shedivat et al., 2018) and (Stadie et al., 2018)"

The notation in Equations 6,7,8,9, the notation for which MDP is sampled could perhaps be better reflected in the probability distributions.

Section 2, notation inconsistencies between L_{\mathcal{M}} and L^{\mathca{M}}



Khetarpal et al. Options of Interest: Temporal Abstraction with Interest Functions, 2020

Klissarov and Precup, Flexible Option Learning, 2021

Kurin et al., In Defense of the Unitary Scalarization for Deep Multi-Task Learning, 2022

Vivek Veeriah, Tom Zahavy, Matteo Hessel, Zhongwen Xu, Junhyuk Oh, Iurii Kemaev, Hado van Hasselt, David Silver, and Satinder Singh. Discovery of Options via Meta-Learned Subgoals. In NeurIPS, 2021.

Xu, Z., van Hasselt, H. P., and Silver, D. Meta-gradient reinforcement learning, 2018

Zahavy et al., A Self-Tuning Actor-Critic Algorithm, 2021

Zheng, Z., Oh, J., and Singh, S. On learning intrinsic rewards for policy gradient methods, 2018

---

> ### Author Response · Authors · 2023-02-07
> **Response to Reviewer shTx**
>
> We thank the reviewer for taking the time to review our paper and providing many suggestions on how to improve the positioning of our paper in the literature. We plan to address the following concerns by making changes to the manuscript:
>
> - Running the algorithms for more seeds
>     - We agree that 3 seeds are too low for reliable results, we will plot the results with 9 seeds in our next revision (we are currently running additional seeds)
> - The amount of pretraining for each method is different…
>     - We ran each algorithm until convergence or roughly to a week of training time, we will make this clear in the main text
>     - We would like to point out that we are showing the performance wrt. number of episodes used for training. However, each algorithm uses different method for optimization and therefore requires less or more data to perform updates. Tthe runtimes of all algorithms are similar:
>         - MLSH uses PPO which performs several updates with the same data (more computation with same amount of data compared to FAMP)
>         - MAML uses TRPO in the outer loop and therefore may need to recompute the objective several times with the same data to satisfy the KL constraint (more computation with same amount of data compared to FAMP)
>         - Since FAMP uses policy gradient for both inner and outer update, it only uses the data once and thus has to resample more often
>     - While MAML may still indeed slightly improve with additional training time according to Figure 8, continuing to 10^6 would take multiple weeks of training time and is not feasible
> - We will add relevant suggested literature about meta-gradients to our related work section
> - Multi-task baseline uses a hierarchical policy and is trained to maximize immediate (0-shot) average return on training tasks. All parts of the policy are optimized during the training. At test time, only high-level policy is fine-tuned to particular task. In hindsight, it may be more appropriate to fine-tune the whole policy during the test time as well.
> - “Learn all” baseline is the ablation of FAMP where all parts of the hierarchical policy are both learned (outer loop) and adjusted in the inner update. It can be seen as using MAML with hierarchical policy. We will make this more explicit in the paper.
> - We will make more clear in the paper why we do not consider certain baselines:
>     - We consider a setting in which the agent does not know which environment it currently is in  (i.e. the agent does not have environment id during training). This information is also not necessary for MLSH but is used by the high-level policy of Veeriah et al. 2021 (their high-level policy selects option based on state and goal).
>     - OC, MOC, IOC (Khetarpal et al. 2020) - and all similar methods rely on a learned Q-function. However, it is not straightforward to imagine how these algorithms should be adapted into a multi-task setting without environment id. A single learned Q-function would not work well for tasks which are different.
>     - An additional problem with using the method of Veeriah et al. 2021 as a baseline is that the source code of the method is not available

---

> > ### Comment · Reviewer_shTx · 2023-02-14
> > **Response to Authors**
> >
> > I am very glad to see the authors address the issue of reproducibility by adding to the number of seeds.
> >
> > I am still confused why running until convergence is the right setup. Wouldn't it be more fair to give each algorithm a set number of pretraining steps? What is intended by giving different algorithms different amounts of training? I feel like not only should this be addressed in the main text, but there should be a figure that clearly illustrates those differences.
> >
> > Thank you for explaining in more depth the baselines. Hopefully this can be reflected in the paper as well.
> >
> > In general I would like to see the promised changes in the paper already, I believe it is possible to do so.
> >
> > I believe this question was not addressed:
> > In the appendix it is said "MAML uses a more complex outer gradient update while FAMP relies on a simple policy gradient." I am not sure I understand what that means exactly, could the authors expand?
> >
> > Concerning MOC and IOC, both of these works do not use a Q function, but instead a value function used as a critic (perhaps this is what was meant?). In this case, I am not sure why learning a single value function would be bad, if a single policy is learned across tasks? Even if the authors do not implement these baselines, I think the authors should rephrase some of the claims of the paper that mention that only IOPG can learn multiple options in parallel, which is not the case.

---

> > > ### Author Response · Authors · 2023-02-16
> > > **Response to Reviewer shTx**
> > >
> > > We’ve added a revision with the following changes based on your comments:
> > >
> > > - Figure 6 now shows 9 seeds for all algorithms, we are currently running more seeds for Taxi experiments
> > > - We’ve made the pre-training length explicit at the start of the experiments section (one week of training or convergence is reached)
> > >     - Our main aim was to give each algorithm a week of training time. However, if they converged earlier we stopped the training. As we've mentioned in our response, each of the algorithms (FAMP, MLSH, MAML) uses different amount of compute/data per epoch (see below for more detailed explanation of MAML), we therefore gave each of them similar amount of compute.
> > > - We’ve added the suggested literature into RW section
> > > - We’ve adjusted multi-task policy in Taxi (allowing it to adapt during test time) and added more thorough discussion about how each baseline works in the experiments section
> > > - We’ve added MOC and IOC to the related work and at the end of section 3
> > >     - We hope that the part in related work now makes it clear that MOC also allows options to be learned at the same time
> > >     - We were referring to the option value function Q(s,o) and Q(s,o,a) in the IOC/MOC algorithm pseudocodes. To make good value estimates in multi-task setting one would need to do goal conditioning and for that one would need to know what the goal is. Our method and MLSH assume that we can sample from the distribution of tasks but that the task id (or the goal) is not given. This would also be the case if IOC/MOC are adapted to only work with value function V(s). We hope that this is now made more clear in the paper (end of sec 3).
> > > - We’ve added discussion about the differences between our method and the one introduced by Veereiah et al., including why it is not applicable in our setting
> > >
> > > As for your question about MAML:
> > > - If we consider the adaptation step (line 13 in Algorithm 1) as inner update and line 15 as outer update, our algorithm uses simple policy gradient in both inner and outer update. MAML uses simple policy gradient for inner update but uses TRPO to perform outer update. It therefore needs to reevaluate the objective several times during the backtracking step of TRPO. In this step the algorithm checks whether KL constraint is satisfied and whether the objective improved. This is also the reason why MAML requires more computation but is more data-efficient. Since the inability to use TRPO in the outer update could be considered a limitation of our method we’ve made this explicit in the discussion section. We hope that this answers your question about MAML.

---

### Review · Reviewer_nzQu · 2023-01-31

**Summary Of Contributions:**

This paper proposes a hierarchical RL architecture that learns a set of option (i.e., sub-policies with their termination functions) end-to-end using meta-gradients in a multi-task setting. The empirical results show that the proposed method outperforms a hierarchical baseline (MLSH) and other non-hierarchical baselines in Taxi and Ant Maze.

**Audience:**

Yes

**Broader Impact Concerns:**

I have no broader impact concern.

**Claims And Evidence:**

No

**Requested Changes:**

- Add two baselines that are mentioned in the weaknesses section.
- Compare against MLSH on more challenging tasks that MLSH was evaluated on.
- Provide insights how the proposed method addresses option collapse or degenerate problem that is mentioned in the weakeesses section.
- Add one of the unlucky seeds back to the result.
- Highlight how similar/different the proposed method is compared to Verriah et al.
- (Recommendation) Explicitly state a list of hypotheses to verify at the start of the experiment section. This is because there are so many baselines to compare and it is not immediately clear what each comparison is showing.

**Strengths And Weaknesses:**

**Strengths**

- Discovering temporal abstractions is one of the most important challenges in RL. This paper proposes a method that is technically reasonable in the sense that the options were discovered with a principled objective: to maximise rewards on the tasks we care about.
- The paper provides a nice summary of relevant background and motivates the problem well.

**Weaknesses**

- The experiments are not comprehensive enough to support the claims of the paper.
  - Overall, it could be much better to explicitly list a set of hypotheses (or questions) that the paper is trying to investigate through the experiments.
  - While this paper claims that their method is better than MLSH, the domains considered in this paper are much simpler compared to the domains in the MLSH paper. While Ant Maze was taken from the MLSH paper, it is solvable using a single-task non-hierarchical policy. MLSH was in fact evaluated on much more challenging tasks such as Ant Obstacle, where the single-task policy completely fails.
  - Some important non-hierarchical baselines are missing. The first one is MAML which is missing in Taxi domain. Another important baseline would be a flat multi-task policy that is pre-trained on the training tasks and fine-tuned on the test tasks at test time. This would justify the importance of hierarchical RL on those domains.
  - The discussion about the role of learned termination functions compared to options with a fixed number of steps is so important that the relevant result should be included in the paper rather than in the appendix.
  - What does prevent your method from collapsing to a single option or converging to trivial options such as primitive actions? While these questions have proven to be difficult to address, this paper does not theoretically or empirically verify how exactly the proposed method addressed these challenges.
  - It seems wrong to drop the result from one of the seeds simply because it was unlucky. I suggest including all 5 seeds for fair comparison between methods or use longer episodes as the paper suggested.

- The contribution and novelty of the work needs to be clarified. While the proposed method is very similar to Veeriah et al.'s work in that they also proposed to use meta-gradients to discover options with the same inner/outer optimisation objective, the departure from Veeriah et al. is not much highlighted throughout the paper. The current presentation could be misleading to the readers who are not familiar with Veeriah et al. Although the problem setting is slightly different, it would be still important to highlight the difference/similarity to Veeriah et al. at least in the method and related work section.

- Clarity
  - "Ours + FT" in the plots are not mentioned or explained in the text.

---

> ### Author Response · Authors · 2023-02-07
> **Response to Reviewer nzQu**
>
> We thank the reviewer for taking the time to review our paper. We will make the following changes to the manuscript according to your recommendations:
>
> - We think you are right that showing the results with the outlier removed makes the results less transparent, we will add a figure with all points including these outliers
> - Moving the discussion about the role of learned termination functions compared to options with a fixed number of steps to the main paper is a great suggestion! We will move it to the main paper.
> - Explicitly state a list of hypotheses to verify at the start of the experiment section
> - Make the differences with Verriah et al. explicit in the paper
>     - The difference between our method and the one used by Veeriah et al. 2021 is that they meta-learn option rewards networks and terminations and adjust both sub-policies and high-level policy in the inner update whereas we rely on the task reward and only adjust high-level policy in the inner update. They also allow the agent to use primitve options (option = action in MDP) during training and test time and use a switching cost to prevent high-level policy from selecting primitive options too often. This is not the case in our work where we do not use primitive options. Lastly, they use both current state and goal (input that encodes which goal the agent should reach in the environment) as an input to the high-level policy. The agent thus knows which task it is currently in. We will highlight these differences in our related work section.
> - Add the discussion about how the proposed method addresses option collapse and degenerate problem:
>     - Although our objective does not explicitly constraint the length of the options to prevent option collapse or options that are too short, the proposed objective implicitly regularizes towards options that have appropriate length in two ways:
>         - If one uses less options than tasks, using multiple options should lead to higher return (averaged across tasks) when compared to using only a single option. This prevents the options from becoming too long and not reusable (option collapse). Note that this is different from methods that are trained in single-task regime such as the Option Critic.
>         - Similarly, optimizing for fast adaptation by keeping the amount of inner updates small, the algorithm is incentivized to use options that are not too short (degenerate 1-step options). This is because options that are too short would not allow high-level policy to fully adapt to new tasks within several inner updates and would lead to lower average return.
>
>  We would also like to further discuss the following two points that you raised:
>
> - MAML baseline in Taxi
>     - Currently, we use “Learn all” ablation to showcase the performance of MAML + hierarchical policy on Taxi. In our opinion, this is a relevant MAML baseline that should give an indication of how well MAML performs.
> - Multi-task policy that is pre-trained on the training tasks and fine-tuned on the test tasks at test time
>     - In taxi we are currently using a hierarchical multi-task baseline that is pre-trained for best-zero shot performance on multiple tasks and whose high-level policy is then fine-tuned during test-time. We will adjust this baseline to fine-tune whole policy during test-time instead of only high-level policy since this would make for a stronger baseline and would come close to the multi-task baseline that you are suggesting.
> - Evaluation on Ant Obstacle from MLSH
>     - We would like to point out that our goal is to learn a hierarchy during execution of the task, whereas the setup in AntObstacle in MLSH relies on “curriculum learning” with sparse reward to scale to larger environment:
>         - Since the agent only receives the reward 1 when reaching a goal PPO can never learn because a sequence of random moves will not lead to the reward.
>         - The orientation of the Ant is reset when high-level policy selects an option
>         - MLSH is first pre-trained on two tasks that require the Ant to move up/right with dense reward (TwoWalk Fig. 6). and subsequently used to solve a task with a sparse reward that requires the ant to move to the top right corner of the environment
>     - Consequently, this setting is in fact testing different capabilities of the method (learning from carefully structured curricula rather than efficiently learning end-to-end in complex tasks) and is not really directly comparable to the settings presented in our work

---

> > ### Comment · Reviewer_nzQu · 2023-02-14
> > **Response to the authors**
> >
> > Thank you for addressing some of my comments.
> > However, I still have a few remaining concerns.
> > * I was pointing out the lack of "non-hierarchical" MAML and multi-task policy baselines. Your response still seems to refer to the hierarchical versions of them. Can you explain why the hierarchical baselines are more appropriate?
> > * Regarding comparison to MLSH, I still believe that both MLSH and your method are proposed to learn reusable options that allow the agent to quickly learn and adapt to new tasks. I agree that Ant Obstacle is designed for curriculum learning. But, isn't it exactly the setting where the ability to quickly learn new tasks is needed (adapting from easier tasks to harder tasks)? Even if the authors' argument is true, what would be the claim you would like to make by comparing against MLSH, if MLSH is designed for different capabilities?
> > * The above two concerns are why I suggested listing hypotheses that the paper is trying to verify through experiments.

---

> > > ### Author Response · Authors · 2023-02-17
> > > **Response to Reviewer nzQu**
> > >
> > > Thank you for the clarification. We’ve added the changes mentioned in our previous post to the paper.
> > > - We apologize for misunderstanding your initial reply. There are several changes between FAMP and MAML: the use of TRPO instead of REINFORCE, the separation of parameters into those updated in the inner-and outer loops, and the hierarchical policy itself.
> > > The use of an hierarchical policy has been compared to flat policies in several earlier papers, and the use of TRPO vs. REINFORCE was not the focus of the current paper. Therefore, we focused on the difference caused by the sub-division of parameters (learn all baseline) in the Taxi domain and not using the adaptation during the training (multi-task baseline). We agree that it would still be interesting to furthermore test the effect of the hierarchical policy itself. We therefore propose to add one more ablation where we will use our algorithm with a flat policy instead (which results in a set-up similar to that used by MAML). We hope that this addresses your concerns.
> > > - We would like to emphasize that our paper is focused on learning hierarchies end-to-end and that the main focus of MLSH is also on learning hierarchies end-to-end from experience. However, MLSH has both experiments with end-to-end learning of hierarchies and experiments with curriculum learning. While the latter are not really relevant for our claims, MLSH is still the most important method to compare to is in the area of end-to-end learning of hierarchies.
> > > - We have adjusted the experiments section by adding questions and motivation for each baseline and comparison according to your suggestion. We hope that this makes the comparisons more clear.

---

> > > > ### Comment · Reviewer_nzQu · 2023-02-24
> > > > **Response to the authors**
> > > >
> > > > Thanks for the update.
> > > >
> > > > * I still believe that showing a flat policy performance is important because it serves as a lower-bound performance, even though the prior work has already shown the advantage of hierarchical policy over flat policies. I believe that the main idea of MAML is not about TRPO vs REINFORCE but rather its meta-learning objective and how to derive (meta-)gradient, which means MAML can be implemented with any policy optimization algorithms.
> > > >
> > > > * I still do not follow the author's argument about end-to-end learning and curriculum learning. The curriculum learning part of AntObstacle is used simply to define a task distribution, in order to ultimately measure how quickly the agent adapts to new tasks. Isn't this exactly the goal of your method? In other words, both MLSH and FAMP are proposed for fast task adaptation with reusable options NOT for curriculum learning. AntObstacle was used to evaluate exactly this aspect by defining a non-stationary task distribution (its curriculum learning part is just a detail). Please let me know if I am missing something.
> > > >
> > > > * Thank you for adding questions in each paragraph. It looks much clearer to me.

---

> > > > > ### Author Response · Authors · 2023-02-28
> > > > > **Response to Reviewer nzQu**
> > > > >
> > > > > Thank you for further clarification.
> > > > >
> > > > > - You are right about both showing an ablation without hierarchy and the main idea of MAML is the meta-gradient objective and not TRPO vs. REINFORCE. We have added a No Hierarchy baseline in Figure 3 to evaluate the performance of a non-hierarchical policy with MAML. We hope that this addresses your concerns.
> > > > > - We would like to further clarify the difference between the setting from MLSH that we consider in our work and the curriculum setting of AntObstacle in MLSH:
> > > > >   - Learning reusable options end-to-end (the setting that we consider in the paper):
> > > > >     - Environments require sequence of options
> > > > >     - The aim is to extract options from experience
> > > > >     - These options should generalize to similar envs (same domain)
> > > > >   - Generalizing behavior from simple tasks (AntObstacle in MLSH):
> > > > >     - Training tasks and setup are careful constructed by the designer to obtain specific options (learn directional movement primitives in TwoWalk MLSH)
> > > > >     - In the case of TwoWalk environments were designed to require a single option
> > > > >     - These options should then generalize to larger composite envs (AntObstacle combines directional primitives)
> > > > > - In our work, we consider the first setting and claim that our method performs better than MLSH in Taxi and AntMaze environment (from MLSH). As for the generalizing behavior from simple tasks, in our work we are not interested in learning option primitives in carefully constructed settings. While we understand that generalization to larger environments is important, generalizing to larger environments and state spaces using options is a hard problem in general. This problem is currently out of scope for our work and requires further study.

---

> > > > > > ### Comment · Reviewer_nzQu · 2023-03-03
> > > > > > **Response to authors**
> > > > > >
> > > > > > Thank you for adding the baselines.
> > > > > > The final response convinced me why the setting in AntObstacle is out of scope for this work.
> > > > > > Since most of my concerns have been addressed, I recommend accepting this paper.

---

### Review · Reviewer_a2xU · 2023-01-31

**Summary Of Contributions:**

The authors present a novel meta-reinforcement learning method that extends MAML with options. They do this by leveraging IOPG to learn both option policies and their terminations in parallel (i.e. even for the options not currently being executing) and in manner compatible with gradient-based meta-learning. They choose to only update the low-level policy in the outer loop, so that the option are agnostic to the current task, but are generally useful across the task distribution. The high-level policy is updated in the inner-loop, with the overall optimization process incentivizing the learning of options that allow for the rapid adaptation of a high-level controller.

The authors evaluate their method on a set of simple gridworld tasks as well as a more complex continuous control domain. These results show that FAMP is generally better than prior approaches (e.g. flat MAML) in both the meta-RL and multitask settings.

**Audience:**

Yes

**Claims And Evidence:**

Yes

**Requested Changes:**

# Needed for acceptance
* Modifying the text to clarify the design decision brought up in W3, and ablating it if feasible
* Reworking Figure 6 and/or adding additional figures to address W2
* Rerunning on the Ant environment with the random orientation resets as discussed in W4 (to make results as compatible to prior work as possible)

# Strengthen the paper
* A pixel-based environment would highlight scalability as per W1.
* Evaluating FAML on some of the tasks from the original MAML paper would ease concerns raised in W1
* An experiment showing how FAMP and MAML scale with the number of training tasks.

**Strengths And Weaknesses:**

# Strengths
1) simple, well motivated approach. One could argue it is "just" the combination of preexisting methods (IOPG + MAML), but this actually increases the ease of replication and the likelihood of adoption by the broader community.
2) Clearly written, with a fairly comprehensive treatment of related works.
3) Very small number of meta-training tasks needed. I generally thought gradient-based meta-RL needed orders of magnitude more tasks to properly generalize. I'd be curious to see a comparison of how FAMP and MAML scale with the scale of the meta-training task set.

# Weaknesses
1) Relatively simple task domains with obvious hierarchical structure. Many prior approaches have shown promising initial results that subsequently fail to scale up. I'm not convinced this method would outperform e.g. RL^2 on a pixel-based environment. Similarly, I'm worried that in tasks without such a clear separation of concerns FAMP might fail to learn anything.
2) Qualitative evaluation could go farther. Figure 6 in particular feels lacking, as it only shows the behavior of a few options. What to all of the options do? Are the all consistently used?
3) Why aren't the high-level controller parameters optimized in the outer-loop as well? It feels wasteful to start every task with a naive high-level policy. Perhaps this could be seen as enforcing more uniform option usage, but I'd like to see that argument made explicit (and ideally show that the alternative is less performant)
4) Changes to the ant environment from prior work seem poorly motivated. Sure, randomly resetting the ant orientation isn't realistic, but nothing about this environment is very realistic -- might as well follow the procedures of prior work to aid in comparison. Especially when comparing to models tuned for that variant.
5) Apart from the failure mode of num options >= num tasks, another possible failure mode might occur when the number of options is too low relative to the complexity of the environment. For example, I have a hard time believe an agent with only 2 task-agnostic options could navigate to an arbitrary 2D location. This suggests a "sweet spot" where you have a enough options to do everything you need to do in the environment but not so many that each ends up just corresponds to a task-specific policy. Would be nice if you address this in the paper (or convince me this 2nd failure mode doesn't exist)

---

> ### Author Response · Authors · 2023-02-07
> **Response to Reviewer a2xU**
>
> We thank the reviewer for taking the time to review our paper and providing helpful suggestions for additional experiments that could strengthen the paper. We would like to provide clarification on your questions and address some issues raised:
>
> - W1:
>     - Our method currently has some limitations wrt. to scaling that we’ve mentioned in the discussion section. The main one is that the computation required to calculate IOPG option responsibilities scales linearly with episode length. These make it hard to scale our approach to larger experiments in its current form. Further work is thus needed to achieve this but we believe that our approach is a relevant step to get there.
> - W2:
>     - In Figure 6, we show option usage across multiple tasks. Cyan/Blue/Purple correspond to the same option in each task (i.e. blue is always option 1, purple option 2 and cyan option 3).
>     - While it is easy to visualize learned options in the Taxi domain because of the finite state space, it becomes difficult in high-dimensional continuous state spaces. We chose this type of visualization (focusing on the 2 most relevant dimensions of the state space) in Ant because we considered it to be the best way to visualize learned options in this high-dimensional state space.
>     - We are not sure that we completely understand your request to adjust this figure since it currently already shows all learned options. Could you please elaborate on how we could improve the qualitative evaluation to address your concern?
> - W3:
>     - We agree that it may feel wasteful to start every task with a naive high-level policy. We indeed consider this choice to be beneficial to ensure exploration in the option space both during the training and test time. Particularly in the Taxi domain (with a sparse reward), meta-learning high-level policy could be detrimental because the algorithm requires some exploration to reach rewards in all corners.
>     - In one of the papers that extended MAML (CAVIA [1] Sec 3.4), authors also showed that using fixed initialization of inner loop parameters made the algorithm more robust to the choice of inner learning rate. Although these experiments were performed on supervised learning domains, we believe that this takeaway is also relevant for RL.
>     - We will make these arguments explicit in the paper.
>     - We are currently running additional seeds for main experiments to address concerns of reviewer shTx and these currently require all of our computation budget. Consequently, we are unsure whether we will be able to run the ablation before submitting the revision. However, we plan to run and add this baseline when these runs are finished.
> - W4:
>     - While we agree that the environment is a simulation and thus may be deemed “not very realistic”, we think that the resets that were present in MLSH warped the environment in the way that is commonly not present in the Mujoco environments. For example, if the ant would fall and lay on its back, it would be “switched” to standing position with fixed orientation when a reset occurs. Additionally, these resets were timed to occur at timesteps when high-level policy chooses an option. We thus think that the removal of the resets makes the environment more realistic and similar to mujoco environments used in RL literature.
>     - Nevertheless, we agree that showing results on original environment can be important to have direct comparison. As we’ve mentioned in the paper, we have ran experiments in which we’ve compared MLSH with our algorithm on the original domain with resets and the results were very similar to the ones presented in the paper. We will add these results to Appendix and refer to them in the main paper.
> - W5:
>     - You are right in thinking that a failure mode that arises when there are not enough options to represent optimal trajectories in all tasks can indeed occur. In such case, depending on the task, the consequences of using “too few” options may be less or more severe. For example, as we’ve discussed in the paper, this happened in the Taxi environment when we used only 2 options. While the performance dropped slightly, the agent still learned to solve all tasks with only 2 options.
>     - We will make this more explicit in the paper.
>
> We hope that these changes will address your main concerns.
>
> [1] Luisa Zintgraf, Kyriacos Shiarli, Vitaly Kurin, Katja Hofmann, and Shimon Whiteson. Fast Context Adap-
> tation via Meta-Learning. In ICML, 2019.

---

> > ### Author Response · Authors · 2023-02-17
> > **Response to Reviewer a2xU**
> >
> > We've added the following changes mentioned above in the revision:
> > - Extended the discussion about a failure mode when there are not enough options (W5)
> > - Added the plot with runs on AntMaze with resets in the Appendix (W4)
> > - Made the argument about using uniform high-level policy explicit in the paper (W3)

---

> > ### Comment · Reviewer_a2xU · 2023-03-02
> > **Thank you for your thoughtful response**
> >
> > Really appreciated your responses.
> >
> > * W1 - totally fine so long as the linear scaling is mentioned in the paper.
> >
> > * W2 - apologies for not making this more concrete. My hope was to either use a set of examples problems such that the whole option set is used (currently looks like only 2 options were really needed), or show one example problem and depict how the agent behavior differs when the different options were clamped for some fixed duration (i.e. as opposed to just following the high level policy). That said, I should've been more responsive to this followup, so I don't expect these changes to be made in time.
> >
> > * W3 - this is really interesting! I definitely support ablating this aspect, but given compute constraints and prior work, an explicit statement of your reasoning seems fine to me.
> >
> > * W4 - I hadn't realized these resets were timed to option period boundaries; I agree this isn't desirable. Ablation still appreciated, but agree that your version should be the new default.
> >
> > * W5 - Thanks!
> >
> > Under the assumption the revisions you mentioned in your response are done and confidence intervals are added, I'd be approving of this works publication.

---

### Decision · Action_Editors · 2023-03-05

**Recommendation:** Accept as is

**Comment:**

Pretty clear cut submission and discussion. There were some initial concerns about the quality of the experiments and design choices at the reviewer's and AE's requests the experiments were significantly improved. The authors also clarified several misunderstandings. Three expert reviewers in the area were all supportive of acceptance!

**Audience:**

The topic of the paper is options, meta-learning and discovery: extremely relevant and interesting to the RL community.

**Claims And Evidence:**

Claims are clearly supported by evidence. As highlighted by the reviewers this work is particularly honest about the limitations of the proposed methods and the authors did a good job adding more seeds and better measure so confidence. Soldi work.